# A Review: Antimicrobial Therapy for Human Pythiosis

**DOI:** 10.3390/antibiotics11040450

**Published:** 2022-03-26

**Authors:** Sadeep Medhasi, Ariya Chindamporn, Navaporn Worasilchai

**Affiliations:** 1Department of Transfusion Medicine and Clinical Microbiology, Faculty of Allied Health Sciences, Chulalongkorn University, Bangkok 10330, Thailand; sadeep.m@chula.ac.th; 2Department of Microbiology, Faculty of Medicine, Chulalongkorn University, Bangkok 10330, Thailand; ariya.c@chula.ac.th; 3Department of Transfusion Medicine and Clinical Microbiology, Faculty of Allied Health Sciences, Immunomodulation of Natural Products Research Group, Chulalongkorn University, Bangkok 10330, Thailand

**Keywords:** human pythiosis, antibacterial, antifungal, immunomodulatory, drug repurposing

## Abstract

Human pythiosis is associated with poor prognosis with significant mortality caused by *Pythium insidiosum*. Antimicrobials’ *in vitro* and *in vivo* results against *P*. *insidiosum* are inconsistent. Although antimicrobials are clinically useful, they are not likely to achieve therapeutic success alone without surgery and immunotherapy. New therapeutic options are therefore needed. This non-exhaustive review discusses the rationale antimicrobial therapy, minimum inhibitory concentrations, and efficacy of antibacterial and antifungal agents against *P. insidiosum*. This review further provides insight into the immunomodulating effects of antimicrobials that can enhance the immune response to infections. Current data support using antimicrobial combination therapy for the pharmacotherapeutic management of human pythiosis. Also, the success or failure of antimicrobial treatment in human pythiosis might depend on the immunomodulatory effects of drugs. The repurposing of existing drugs is a safe strategy for anti-*P. insidiosum* drug discovery. To improve patient outcomes in pythiosis, we suggest further research and a deeper understanding of *P. insidiosum* virulence factors, host immune response, and host immune system modification by antimicrobials.

## 1. Introduction

Human pythiosis is an infectious disease with high morbidity and mortality [1]. *Pythium insidiosum*, a fungus-like aquatic oomycete microorganism, is a causative agent of pythiosis. The motile flagellate zoospore plays a significant role in initiating an infection. The zoospores of *P. insidiosum* adhere to the skin cut or wound sites and encyst on the surface of the injured tissue(s). The encysted spore develops a germination tube (hypha) that uses chemotaxis to find the host and infiltrate human blood vessels [1,2]. Pythiosis risk is higher in tropical and subtropical regions, including Southeast Asia, eastern coastal Australia, and South America [3]. 

Human pythiosis is associated with a poor prognosis due to the difficulties in diagnosing the infection and the lack of effective therapeutic agents against this disease [4]. The clinical features of human pythiosis are classified into four forms: (i) vascular pythiosis characterized by arteritis, thrombosis, gangrene, aneurysm, or limb claudication; (ii) ocular pythiosis characterized by corneal ulcers, decreased visual acuity, conjunctival redness, eyelid swelling, or multiple, linear, tentacle-like infiltrates and dot-like or pinhead-shaped infiltrates in the surrounding cornea; (iii) cutaneous and subcutaneous pythiosis characterized by a granulomatous and ulcerating lesion in the face or limbs, cellulitis, soft tissue abscess, or lymphadenopathy; and (iv) disseminated pythiosis characterized by the infection of internal organs [1,4,5]. The risk factors for vascular pythiosis include thalassemia, hemoglobinopathy, paroxysmal nocturnal hemoglobinuria, aplastic anemia, and leukemia because *P. insidiosum* has a higher affinity for iron [6,7].

When an infection is diagnosed as *P. insidiosum*, the therapeutic options include surgery, pharmacotherapy, and immunotherapy (Figure 1) [8]. Surgical intervention is the mainstay treatment for managing human pythiosis, but such treatment substantially increases the financial burden on patients, postsurgical complications, and uncontrolled infection [9]. Immunotherapy is a promising approach for human pythiosis treatment where antigens of *P. insidiosum* from *in vitro* cultures are injected into the patient [10,11]. The mechanism behind *P. insidiosum* antigen (PIA) immunotherapy in human pythiosis includes a switching from the host’s T helper-2 (Th2) to T helper-1 (Th1) mediated immune response in the host; the Th1 response producing higher levels of interferon-γ (IFN-γ) and interleukin 2 (IL-2) [10,12,13]. Even though a good prognosis in PIA-treated patients can be implied by Th2 to Th1 switching, the efficacy of *P. insidiosum* antigen is inconclusive when used as immunotherapy in human pythiosis [7,12,14]. *In vitro* studies have demonstrated the anti-*P. insidiosum* effect of antifungals even though the *P. insidiosum* lacks the antifungal drug-target: ergosterol biosynthetic pathway [15]. However, a significant concern with antifungals is the contradictory results in susceptibility to *P. insidiosum* in *in vitro* and clinical use [15,16].

This review focuses on evidence supporting and disputing the effectiveness of antimicrobials to expand the pharmacotherapeutic role of antimicrobials in the management of human pythiosis. We do not explicitly discuss the biology of *P. insidiosum*, the pathogenesis of the *P. insidiosum* infection in humans, or the management of human pythiosis with immunotherapy and surgical intervention. Finally, we conclude with general remarks on future strategic options for managing human pythiosis.

## 2. Principles of Antimicrobial Therapy

Antimicrobial therapy should achieve a clinical response by eliminating the invading microorganism(s) while minimizing cost, adverse effects, and antimicrobial resistance [17,18]. When selecting appropriate antimicrobial therapy, both pharmacokinetic and pharmacodynamic properties of the drug(s) must be considered to ensure that effective agents are administered in sufficient doses for therapeutic success [19]. For species such as *P. insidiosum*, identifying potential targets for antimicrobials is necessary for managing pythiosis. The microbial cell wall is a critical target for antimicrobials, and the cell wall of *P. insidiosum* is primarily composed of β-glucan and cellulose [20]. However, the cell wall of *P. insidiosum* lowers the penetration of drug molecules and prevents drug access to targets inside the cell wall [21]. The gene expression of cytochrome oxidase 2 (*COX2*) in Thai *P. insidiosum* strains was 2.5-fold higher at 37 °C compared to the expression at 27 °C [22]. In addition, the elicitin protein, *ELI025*, was highly up-regulated in *P. insidiosum* hyphae at 37 °C compared to hyphae grown at 28 °C and facilitated the evasion of the host antibody response [23]. *COX2* and *ELIO25* can be candidate targets for controlling *P. insidiosum* infection.

Several antifungal and antibacterial drugs have been examined for their susceptibility profile against *P. insidiosum* in an *in vitro* study (Figure 2). They have been tried to manage human pythiosis but have been successful only in a few cases [4]. *P. insidiosum* keratitis was successfully managed in a 20-year-old Japanese man following triple antibiotic therapy (minocycline ointment four times a day, chloramphenicol eye drops hourly, and linezolid 1200 mg orally twice a day) [24]. Recently, a *P. insidiosum* keratitis patient was successfully managed with topical 0.2% linezolid and topical 1% azithromycin, administered hourly [25]. Antimicrobial susceptibility testing (AST) is a procedure to determine the concentration of an antimicrobial that inhibits microbial growth *in vitro* by establishing minimum inhibitory concentration (MIC), which is the lowest concentration of an antimicrobial that inhibits visible growth of a microorganism [26,27]. Table 1 summarizes the methods used to determine the MIC of antimicrobial drugs against *P. insidiosum* discussed in our review.

## 3. Why Do Antimicrobial Treatments Fail?

Factors contributing to the antimicrobial treatment failure include antimicrobial agent’s pharmacokinetic and pharmacodynamic issues related to the antimicrobial agent, lack of pathogen control, development of infection complications, drug-resistant pathogens, conflicting AST results, disparities between *in vitro* and *in vivo* efficacy, host immune response, and wrong choice of antimicrobial drug (Figure 3) [33,34]. Pharmacokinetics variability can be defined as differences in plasma antimicrobial exposure, impacting treatment success [35]. Antimicrobials, like beta-lactams and aminoglycosides, achieve suboptimal plasma concentrations in critically ill patients due to increased volume of distribution and increased renal and hepatic clearance [36,37]. As another example, linezolid’s pharmacokinetic variability results in adverse effects and ineffective therapy because of the narrow therapeutic window of linezolid [38].

*P. insidiosum* produces six enzymes (ERG3, ERG5, ERG11, ERG20, ERG24, and ERG26) included in the sterol biosynthetic pathways [16]. However, more than 40 enzymes are involved in the sterol biosynthetic pathways; thus, drugs targeting sterol pathways exhibit limited efficacy against *P. insidiosum*. These drugs cannot be exploited for rationalized and successful management of pythiosis [39]. Different strategies could be considered to prevent the antimicrobial treatment failure in pythiosis, namely: delivering adequate concentration of antimicrobial drug at the site of infection [40], increased periods of exposure of *P. insidiosum* to the antimicrobial drug [41], redesigning drug to penetrate the outer membrane of *P. insidiosum* and avoid being pumped out of the membrane [42], and modulate host immunity [43].

## 4. Immune Response and Antimicrobial Therapy

The innate immune system protects the host from various toxins and infectious agents, including bacteria, fungi, viruses, and parasites via phagocytosis and intracellular killing, recruitment of other inflammatory cells, and presentation of antigens [44]. The innate immune system is highly complex and comprises physical and anatomical barriers, effector cells, antimicrobial peptides, soluble mediators, and cell receptors [45]. However, pathogens can breach the early innate immune mechanisms. In these circumstances, a strategy to modify the function of immune cells can lead to the elimination of the pathogenic intruder [46]. Interestingly, host immunity is often overlooked in the process of pathogen clearance. A favorable innate immune response can considerably reduce the need for more prolonged antimicrobial therapy in infections [47]. 

Once *P*. *insidiosum* enters and adheres to the host tissues, the soluble exoantigens from *P*. *insidiosum* trigger the Th2 response and lock the host immune system into a Th2 subset. Further, *P*. *insidiosum* protects itself from the host immune system by concealing inside the eosinophilic material formed by the eosinophil degranulation, which helps protect the *P*. *insidiosum* from being fully presented to the host’s immune system [10]. Toll-like receptors (TLRs) play a central role in the innate immune system by recognizing pathogen-associated molecular patterns and triggering downstream signaling pathways that activate the innate immune response [48]. Wongprompitak et al. demonstrated that both zoospores and hyphae of *P*. *insidiosum* induced a TLR2-mediated innate immune response with a subsequent increase in the levels of the pro-inflammatory cytokines IL-6 and IL-8 [49]. 

To combat the pathogen and prevent its spread, it is rational to administer antimicrobial drugs that interact with the host’s innate immune system to provide profound indirect effects and enhance pathogen clearance. Antimicrobial drugs have been shown to modify the immune responses to infection, guiding improved treatment strategies in human pythiosis (Table 2 and Figure 3).

## 5. Antibacterial Drugs against *P. insidiosum*

Previous *in vitro* screening of antibacterial drugs has identified tetracycline, minocycline, tigecycline, azithromycin, clarithromycin, erythromycin, gentamicin, streptomycin, paromomycin, neomycin, linezolid, nitrofurantoin, quinupristin-dalfopristin, chloramphenicol, clindamycin, and mupirocin, which demonstrated inhibitory activity against *P. insidiosum* [29,30,31,63,64,65]. Among the arsenal of antibiotics, the best-studied antibiotics in human pythiosis are tetracyclines, macrolides, and oxazolidinones. This section discusses different classes of antibacterial drugs to manage human pythiosis.

### 5.1. Tetracyclines

Tetracycline antibiotics such as tetracycline, tigecycline, and minocycline inhibit bacterial protein synthesis by binding with the bacterial 30S ribosomal subunit [66]. Tetracyclines can inhibit mammalian collagenase activity and assist wound healing [67]. Further, tetracyclines potentiate the innate immune response and augment the resolution of inflammation [50].

Based on the *in vivo* studies in rabbits, minocycline in combination with immunotherapy may be an effective therapeutic medical treatment of pythiosis to heal injuries [68]. Worasilchai et al. evaluated the *in vitro* susceptibility of human, environmental, and animal *P. insidiosum* isolates to eight antibiotic classes and demonstrated that tetracyclines and macrolides inhibited the *in vitro* growth of *P. insidiosum* isolates at concentrations 10 to 100 times lower than those observed for previously studied antifungal drugs [28]. Also, the combination of tetracyclines and macrolides resulted in a synergistic effect that reduced MICs against *P. insidiosum* isolates. Loreto et al. also reported a similar *in vitro* susceptibility of *P. insidiosum* isolates to tetracyclines and their superior potency compared to amphotericin B, echinocandins, and triazole antifungals [29].

### 5.2. Macrolides

Macrolides are the group of antibiotics that inhibit bacterial protein synthesis by binding with the bacterial 50S ribosomal subunit. Common macrolides include erythromycin, clarithromycin, and azithromycin [69]. Among the macrolides, azithromycin, in particular, is highly accumulated in phagocytes and is targeted to the sites of infection [70]. Azithromycin reduces the production of IL-12, resulting in enhanced Th2 response [51]. Th2 cells are involved in wound healing and tissue repair [71,72]. The immunomodulatory activities of macrolides are evident with both pro-inflammatory and anti-inflammatory effects. For example, erythromycin can suppress pro-inflammatory cytokine production, such as IL-6, IL-8, and tumor necrosis factor-α (TNF-α) [73].

Jesus et al. investigated the antimicrobial activity of azithromycin alone and in combination with minocycline against *P. insidiosum* in a rabbit model [74]. The results revealed a strong *in vivo* activity of azithromycin (20 mg/kg/day twice daily) alone and combination with minocycline (10 mg/kg/day twice daily) against subcutaneous lesions. In an *in vitro* susceptibility study, the MICs of azithromycin and clarithromycin were less than 4 μg/mL for *P. insidiosum* isolates [29].

### 5.3. Oxazolidinones

Oxazolidinones such as linezolid inhibit bacterial protein synthesis by binding with the 50S subunit of the ribosome [75]. The suppression of the synthesis of pro-inflammatory cytokines, such as interleukin-1β (IL-1β), IL-6, IL-8, IFN-γ, and TNF-α by linezolid has highlighted an exciting role of linezolid in immunomodulatory effects [52,53,54]. Linezolid may significantly reduce the inflammatory damage induced by the excessive release of pro-inflammatory cytokines during critical infections [76].

In a rabbit model of *P. insidiosum* keratitis, topical linezolid demonstrated superior efficacy and safety compared to azithromycin and tigecycline after prolonged treatment for more than 3–4 weeks [77]. 

### 5.4. Lincosamides, Streptogramins, and Phenicols

Lincosamides, streptogramins, and phenicols inhibit bacterial protein synthesis by interacting with the 50S subunit of bacterial ribosomes [78]. Among lincosamides, clindamycin possesses immunomodulatory activity by suppressing the release of inflammatory cytokines such as TNF-α and IL-1β and enhancing the phagocytosis of microorganisms by host cells [55,56]. Quinupristin and dalfopristin, used in a fixed combination, belong to a class of streptogramins [78]. Quinupristin-dalfopristin decreased the concentration of pro-inflammatory cell wall components (lipoteichoic acid and teichoic acid) and TNF activity in cerebrospinal fluid compared to the ceftriaxone-treated rabbits [57]. A previous report showed that chloramphenicol, a member of the phenicols group, elevated the IL-10 levels, a potent anti-inflammatory cytokine [58].

Lincosamides, streptogramins, and phenicols have shown the ability to inhibit the growth of *P. insidiosum* isolates. The microdilution-based MIC ranges (with geometric means) of lincosamides, streptogramins, and phenicols against *P. insidiosum* were reported to be 2 to >4 μg/mL, 1 to >2 μg/mL, and 8 to >16 μg/mL, respectively [30].

### 5.5. Aminoglycosides

Aminoglycosides such as gentamicin, streptomycin, paromomycin, and neomycin bind to the bacterial ribosome and inhibit protein synthesis [79]. Streptomycin stimulated the *in vitro* growth of one of the Thai *P. insidiosum* isolates [80]. Aminoglycoside antibiotics inhibited the *in vitro* growth of *P. insidiosum*; however, they may not be clinically relevant due to the high MIC values [31]. Therefore, aminoglycosides for clinical use in managing human pythiosis are questionable.

### 5.6. Miscellaneous Antibacterial Drugs

Nitrofurantoin is used to treat urinary tract infections and works by attacking bacterial ribosomal proteins non-specifically, causing complete inhibition of protein synthesis [81]. *P. insidiosum* mycelial growth was inhibited with nitrofurantoin (MIC range of 64 to >64 μg/mL) in an *in vitro* susceptibility test [30]. 

Mupirocin inhibits bacterial protein and RNA synthesis by reversibly inhibiting isoleucyl-transfer RNA [82]. A study evaluating the *in vitro* susceptibility of Brazilian *P. insidiosum* strains showed that mupirocin could inhibit the growth of *P. insidiosum* isolates at MIC lower than 4 μg/mL [29]. 

## 6. Antifungal Drugs against *P. insidiosum*

Studies have focused on several antifungal medications, such as polyenes, azoles, allylamines, and echinocandins, for the adjunctive therapy in managing human pythiosis [4]. Despite the evidence of anti-*P. insidiosum* effects, it has been highly challenging to achieve consistently effective antifungal treatment in human pythiosis. 

### 6.1. Polyenes

Amphotericin B is a polyene antifungal that binds to ergosterol in the fungal cell membrane, which alters cell membrane permeability leading to the loss of intracellular components [83]. Two Australian cases with subcutaneous pythiosis responded well to amphotericin B treatment [84]. However, the evidence of the effectiveness of amphotericin B against other forms of human pythiosis and substantial activity against *P. insidiosum* is lacking [14,85,86,87]. 

Studies have shown amphotericin B’s immunomodulatory properties, which activate the host’s innate immunity [88]. Nitric oxide (NO) is an endogenous regulator of inflammation and an antibacterial agent, and it plays a crucial role in wound repair [89,90]. Amphotericin B can augment the IL-1β-induced inducible nitric-oxide synthase (iNOS) expression and NO production [59]. In addition, amphotericin B is reported to induce oxidative stress and improve antifungal efficacy [91,92].

### 6.2. Allylamines and Azoles

The primary mode of action of allylamines, such as terbinafine, is the inhibition of the enzyme squalene monooxygenase. Therefore, these drugs inhibit the fungal synthesis of ergosterol [93]. Azoles, such as miconazole, ketoconazole, fluconazole, itraconazole, posaconazole, and voriconazole, exhibit antifungal activity by inhibiting the 14α-lanosterol demethylase, a key enzyme in ergosterol biosynthesis, in fungi [94,95]. Studies have suggested that the enhanced microbiocidal activity of monocytes, macrophages, and neutrophils against intracellular *Candida albicans* is enhanced when combined with azoles [61,62]. However, terbinafine has been reported to stimulate pro-inflammatory cytokines [60].

Susaengrat et al. reported favorable responses to voriconazole and itraconazole in Thai vascular pythiosis patients [96]. Synergistic effects have been demonstrated for terbinafine and fluconazole against *P. insidiosum* isolates *in vitro* [97]. A synergistic combination of itraconazole and terbinafine was effective during the *in vitro* susceptibility testing of a *P. insidiosum* isolate from the 2-year-old patient with a deeply invasive facial infection [98]. The growth of *P. insidiosum* isolates was inhibited by terbinafine, and the efficacy of terbinafine increased against *P. insidiosum* isolates when combined with cetrimide, an antiseptic [63]. Pediatricians used a combination of itraconazole and terbinafine to manage a child with vascular pythiosis [99]. *In vitro* susceptibility testing of *P. insidiosum* showed a MICs from 0.5 to 128 μg/mL for terbinafine, 2 to 32 μg/mL for miconazole, 4 to 64 μg/mL for ketoconazole, 1 to >128 μg/mL for itraconazole, 2 to >16 μg/mL for voriconazole, greater than 1 to >32 μg/mL for fluconazole, and >8 μg/mL for posaconazole based on the strains of *P. insidiosum* [9,14].

### 6.3. Echinocandins

Echinocandins, such as caspofungin, anidulafungin, and micafungin, act by inhibiting beta-(1,3)-D-glucan synthase, an enzyme that is necessary for the synthesis of beta-(1,3)-D-glucan, which is an essential component of the fungal cell wall [99]. Studies have documented the immunomodulatory effects of echinocandins with increased fungal beta-(1,3)-D-glucan exposure and caspofungin-induced neutrophil-mediated fungal damage and anidulafungin- and micafungin-induced phagocyte-mediated fungal damage [100,101].

Synergistic anti-*P*. *insidiosum* effects were observed with caspofungin and terbinafine *in vitro* [97]. The MICs of caspofungin and anidulafungin against human *P*. *insidiosum* isolates ranged from 2 to 8 μg/mL [32]. However, when used alone, echinocandins showed poor *in vitro* and *in vivo* activity against *P*. *insidiosum* [29,102]. Caspofungin demonstrated less fungistatic activity against *P. insidiosum* [103]. 

### 6.4. Miscellaneous Antifungal Drugs

Amorolfine, a morpholine derivative, inhibits fungal ergosterol biosynthesis and leads to changes in the membrane permeability, which in turn causes fungal growth inhibition and cell death [104]. Only recently, amorolfine hydrochloride exhibited *in vitro* inhibitory activity against *P. insidiosum* [105]. The MICs of amorolfine hydrochloride tested against *P. insidiosum* isolates were 16 to 64 mg/L. Further, amorolfine hydrochloride produced alterations in *P. insidiosum* hyphae, with changes in the surface of hyphae, intracellular organelles, the cell wall, and plasma membrane of *P. insidiosum*.

## 7. Repurposing Antimicrobials against *P. insidiosum*

Due to the limited success of pharmacological interventions against *P. insidiosum* in humans, identifying novel therapeutic strategies is required to treat *P. insidiosum* infection in humans. Drug repurposing is a process for identifying new therapeutic indications different from the scope of the initial pharmacological indication [106]. For example, antibiotics such as macrolides, tetracyclines, and fluoroquinolones have been used in the clinical management of coronavirus disease 2019 (COVID-19) [107]. Using the drug repurposing strategy, existing FDA-approved antimicrobials can forgo early phases of drug development in managing human pythiosis [108]. Disulfiram irreversibly inhibits aldehyde dehydrogenase (ALDH1A1) and is an alcohol-deterrent medication that causes a severe adverse reaction when patients use alcohol. Disulfiram effectively treats individuals dependent on alcohol but highly motivated to discontinue alcohol use [109]. Krajaejun et al. evaluated disulfiram for its anti-*P. insidiosum* activity using agar- and broth-based methods and revealed that *P. insidiosum* strains were susceptible to disulfiram with MICs ranging from 8 to 32 mg/Liter [110]. Further, disulfiram was found to bind and inactivate aldehyde dehydrogenase and urease of *P. insidiosum*.

Researchers utilize computational and experimental approaches to identify the promising candidates in the drug repurposing process [111]. The computational system uses various databases and computational tools, such as Gene Signature Database (GeneSigDB), Gene Set Enrichment Analysis (GSEA), The Pharmacogenetics and Pharmacogenomics Knowledge Base (PharmGKB), DrugBank, ChemBank, Genecard, Online Mendelian Inheritance in Man (OMIM), PubMed, e-Drug3D, DrugPredict, Promiscuous, Mantra2.0, Protein Data Bank (PDB), DRAR-CPI, repoDB, Repurpose DB, DeSigN, Cmap, and DPDR-CPI, etc. [106,112]. Computational techniques employed for drug repurposing include (i) profile-based drug repositioning, (ii) network-based drug repositioning, and (iii) data-based drug repositioning [113]. Experimental-based approaches validate the computer-generated hits for preclinical drug evaluation [112]. An experimental technique for drug repurposing involves protein target-based and cell/organism-based screens in *in vitro* and *in vivo* assays [114].

Using combination regimens of antibacterial plus antifungal or antibacterial plus antibacterial to achieve synergistic activity is one of the drug repurposing strategies against *P. insidiosum* [115]. Synergism between antibacterial and antifungal against *P. insidiosum* was observed for *in vitro* minocycline with amphotericin B, itraconazole, and micafungin and clarithromycin with micafungin [65]. Susaengrat et al. reported two cases of relapsed vascular pythiosis patients who were successfully clinically managed with a combination of antibacterial plus antifungal [96]. However, isolate-specific combinations for treatment must be implemented because of the varying effectiveness of any given drug combination for different isolates of *P. insidiosum* [116]. Studies have found the enhanced killing effects of multiple classes of antibiotics when combined with NO [117,118]. We expect that NO-containing antibiotics might improve the therapeutic outcomes in patients with pythiosis.

## 8. Conclusions and Future Perspectives

Evidence supports using the antimicrobials reviewed in our article as a new therapeutic option in treating human pythiosis. *In vitro* studies have demonstrated the tetracyclines, macrolides, oxazolidinones, lincosamides, streptogramins, phenicols, aminoglycosides, polyenes, allylamines, azoles, and echinocandins reviewed in our papers inhibit the growth of *P. insidiosum* and have the potential implications for further research on their use in the management of human pythiosis. However, prolonged use of antimicrobials and prolonged treatment with antimicrobials is not warranted due to the side effects and threat of antimicrobial resistance. A practical pharmacological intervention guideline for human pythiosis remains to be discovered and is necessary to assist practitioner and patient decisions, lower treatment costs, and optimize patient outcomes. Despite the disease affecting the most vulnerable populations with higher mortality rates, pythiosis is not included in the Sanford Guide, which provides evidence-based recommendations for treating infectious diseases [119].

In the future, human pythiosis could be managed with antimicrobials owing to their anti-inflammatory and immunomodulatory activities. Clinicians can optimize drug combinations based on the anti-*P. insidiosum* susceptibility testing for the management of pythiosis. Studies have shown the growth inhibitory effects of antimicrobials against *P. insidiosum*; nevertheless, studies regarding the mechanism of action of the antimicrobials against *P. insidiosum* are vital for clinical approval. Researchers must consider the pharmacodynamics principle involved in selecting the antimicrobials to assess the anti-*P. insidiosum* activity. 

Microbial virulence factors are molecules produced by microorganisms and may cause disease in the host (e.g., toxins, enzymes, exopolysaccharides, lipopolysaccharides, lipoproteins, etc.) [22]. The potential virulence factors of *P. insidiosum* include glucan 1,3-beta-glucosidase, heat shock protein (Hsp) 70, and enolase [23]. Keeratijarut et al. reported genetic, immunological, and biochemical characteristics of Exo-1,3-β-glucanase (Exo1) in *P. insidiosum* and found up-regulated *exo1* expression at 37 °C compared to 28 °C, thus suggesting a drug target against *P. insidiosum* [120]. A new therapeutic approach with anti-virulence therapy combined with antimicrobials might prevent the pathogenesis of *P. insidiosum* and limit host damage. Metabolites have been isolated from *Pseudomonas stutzeri* and *Klebsiella pneumoniaei,* and these organisms have shown anti-*P. insidiosum* activity [15,121]. Therefore, the role of potential microbial metabolites in the treatment of pythiosis must be subjected to intense research in the future.

With the evidence of the effectiveness of some antimicrobials in the management of human pythiosis, we suggest using new drug delivery systems to release the drug to the target site in the body and minimize the off-target accumulation of the drug. Antibiotics can be reformulated using nanotechnology-derived delivery systems to improve the targeting and specificity at the infected areas [122]. Due to the genetic variability among individuals, not all individuals with pythiosis exhibit similar therapeutics responses to antimicrobials [123]. Therefore, it is essential to incorporate the pharmacogenomics assay into the clinics to personalize antimicrobial treatment in pythiosis.

## Figures and Tables

**Figure 1 antibiotics-11-00450-f001:**
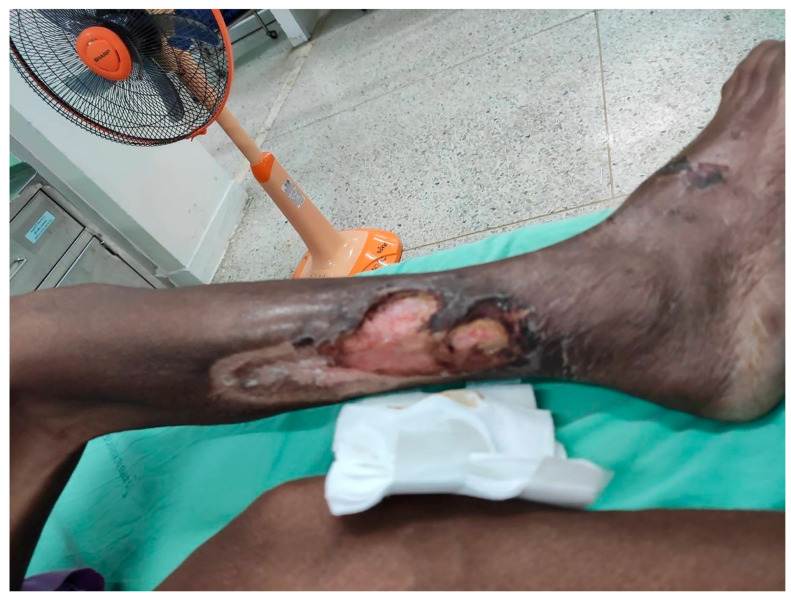
Photograph of human pythiosis. A 46-year-old Thai male with thalassemia was diagnosed with vascular pythiosis. CTA showed the occlusion of the right aorta, and ELISA showed the positive IgG against *P. insidiosum* (with permission). Abbreviations: CTA, computed tomography angiography; ELISA, enzyme-linked immunosorbent assay.

**Figure 2 antibiotics-11-00450-f002:**
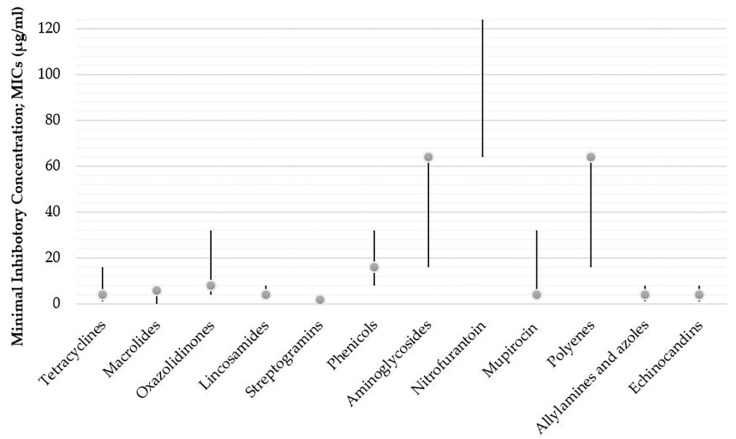
Mode of MIC value of each antibacterial/antifungal class against *P. insidiosum* isolates reviewed in previous publications. Mode of MIC value of *P. insidiosum* isolates against antimicrobial drugs in class different antimicrobial classes: tetracyclines (4 μg/mL) [28,29], macrolides (6 μg/mL) [29], oxazolidinones (8 μg/mL) [29], lincosamides (4 μg/mL) [30], streptogramins (2 μg/mL) [30], phenicols (16 μg/mL) [30], aminoglycosides (64 μg/mL) [31], nitrofurantoin (no data) [30], mupirocin (4 μg/mL) [29], polyenes (64 μg/mL) [29], allylamines and azoles (4 μg/mL) [9], and echinocandins (4 μg/mL) [32].

**Figure 3 antibiotics-11-00450-f003:**
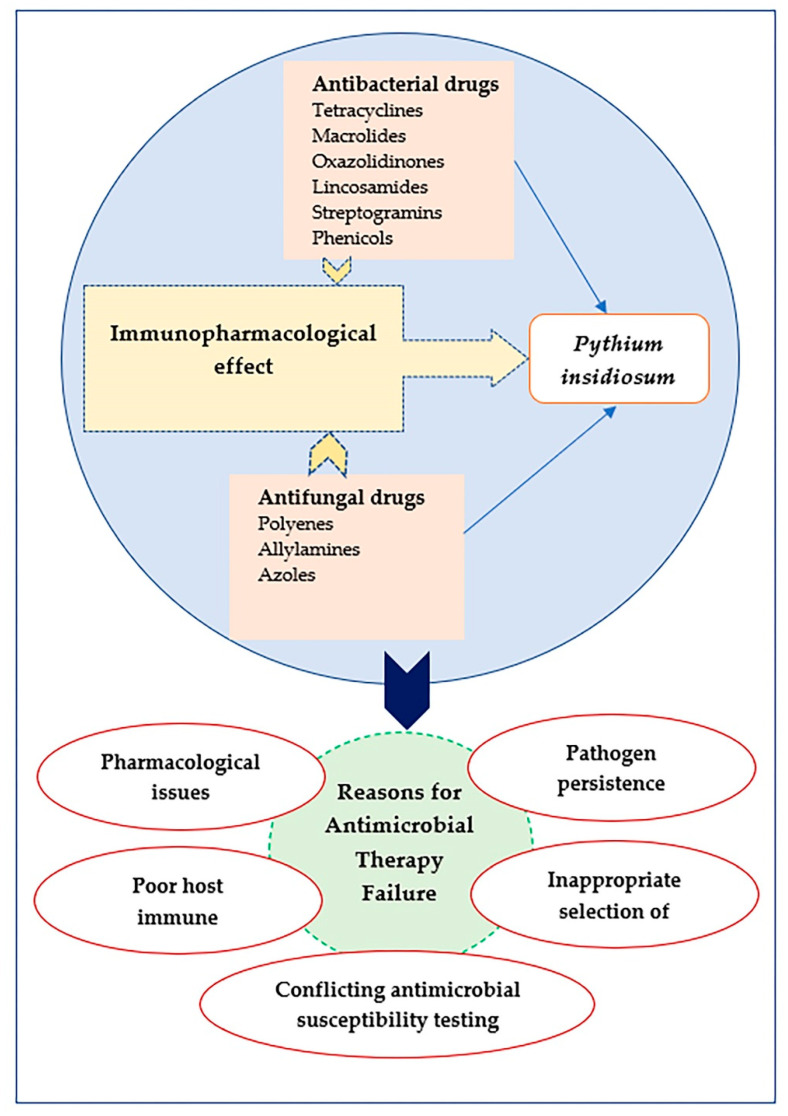
Antimicrobial treatment in the management of *P*. *insidiosum* infection. Antibacterial and antifungal drugs exhibit immunomodulation activity and can improve treatment strategies for human pythiosis. Several mechanisms contribute to antimicrobial failure during the treatment of diseases.

**Table 1 antibiotics-11-00450-t001:** Summary of methods for determining MICs of antimicrobial drugs against *P. insidiosum*.

Antimicrobial Class	Drug	MIC Determination Method(s)	Reference(s)
Tetracyclines	Tetracycline	Broth microdilution	[28]
	Tigecycline	Broth microdilution, disk diffusion, and Etest	[28,29,31]
	Minocycline	Broth microdilution, disk diffusion, and Etest	[28,29]
Macrolides	Azithromycin	Broth microdilution, disk diffusion, and Etest	[28,29]
	Clarithromycin	Broth microdilution, disk diffusion, and Etest	[28,29]
Oxazolidinones	Linezolid	Broth microdilution, disk diffusion, and Etest	[29]
Lincosamides	Clindamycin	Broth dilution	[30]
Streptogramins	Quinupristin and dalfopristin	Broth dilution	[30]
Phenicols	Chloramphenicol	Broth dilution	[30]
Aminoglycosides	Gentamicin	Broth microdilution	[31]
	Neomycin	Broth microdilution	[31]
	Paromomycin	Broth microdilution	[31]
	Streptomycin	Broth microdilution	[31]
	Nitrofurantoin	Broth dilution	[30]
	Mupirocin	Broth microdilution, disk diffusion, and Etest	[29]
Polyenes	Amphotericin B	Etest	[29]
Allylamines	Terbinafine	Broth dilution and radial growth	[9]
Azoles	Miconazole	Broth microdilution	[9]
	Ketoconazole	Broth microdilution	[9]
	Fluconazole	Broth microdilution and agar diffusion	[9]
	Itraconazole	Broth microdilution, radial growth, and agar diffusion	[9]
	Posaconazole	Broth microdilution and agar diffusion	[9]
	Voriconazole	Broth microdilution, radial growth, and agar diffusion	[9]
Echinocandins	Caspofungin	Broth dilution	[32]
	Anidulafungin	Broth dilution	[32]

Abbreviations: MIC, minimal inhibitory concentration.

**Table 2 antibiotics-11-00450-t002:** Immunomodulatory effects of antimicrobials.

Antimicrobial Class	Drug	Immunopharmacological Effect	Reference(s)
Tetracyclines	Tigecycline, minocycline	Potentiate the innate immune response and augment resolution of inflammation	[50]
Macrolides	Azithromycin	Reduce the production of IL-12, resulting in enhanced Th2 response	[51]
Oxazolidinones	Linezolid	Suppress synthesis of proinflammatory cytokines, such as interleukin-1β (IL-1β), IL-6, IL-8, interferon-γ (IFN-γ), and tumor necrosis factor-α (TNF-α)	[52,53,54]
Lincosamides	Clindamycin	Suppress the release of inflammatory cytokines such as TNF-α and IL-1β and enhance the phagocytosis of microorganisms by host cells	[55,56]
Streptogramins	Quinupristin-dalfopristin	Decrease the concentration of pro-inflammatory cell wall components (lipoteichoic acid and teichoic acid) and the activity of TNF	[57]
Phenicols	Chloramphenicol	Elevate the anti-inflammatory IL-10 levels	[58]
Polyenes	Amphotericin B	Activate the host’s innate immunity and augment the IL-1β-induced inducible nitric-oxide synthase (iNOS) expression and the production of nitric oxide (NO)	[59]
Allylamines	Terbinafine	Stimulate proinflammatory cytokines	[60]
Azoles	Fluconazole, voriconazole	Enhance microbicidal activity of monocytes, macrophages, and neutrophils	[61,62]

## Data Availability

Not applicable.

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
