# Peer review of "A Review: Antimicrobial Therapy for Human Pythiosis"

_antibiotics, 2022, doi:10.3390/antibiotics11040450_

Round 1
Reviewer 1 Report
The present article “A review: antimicrobial therapy for human pythiosis” written by Medhasi S. et al, is an important up-to-date article regarding the latest findings on the treatment of human pythiosis.
I find the present article to be well written, documented and very helpful for the scientific community who are dealing with this pathology. The structure of the article follows a good logic and the information is easy to understand. Therefore, I recommend its publication.
Here are some suggestions for improvement:
- Please put all your references in square brackets.
- Please include references for all the information provided in Table 1
- Please include a figure which summarizes the main information of the present review.
Author Response
We appreciate you for your precious time and effort in reviewing our manuscript and providing valuable comments to our paper. Below, we provide the point-by-point responses.Comment 1: Please put all your references in square brackets.
Response: Thank you for pointing this out. We have placed references in square brackets following the ACS style.
Comment 2: Please include references for all the information provided in Table 1.
Response: In the revised manuscript, Table 1 has been renumbered as Table 2. We have included the relevant references in Table 2.
Comment 3: Please include a figure which summarizes the main information of the present review.
Response: We are grateful for this comment as it helps the reader understand the content easily. We have included a figure summarizing the main information of the present review (See Figure 2)
Reviewer 2 Report
Dear Authors,
an element that I miss in this work is the lack of a chapter on methods for determining the MIC value for individual susbtances. Summaries (for example in the form of a table) of how the authors of the publications from which you derived this review determined the MIC values was very interesting and would add value to the manuscript.
In the described cases, the potential sources of the pathogen were identified? As pythiosis happens in animals, could they be a vector?
Please see the requirements for citing works by the journal. It is different from what you have in your manuscript.
Please format the text and table as required by the journal.
Author Response
We appreciate you for your precious time and effort in reviewing our manuscript and providing valuable comments to our paper. Below, we provide the point-by-point responses.Comment 1: An element that I miss in this work is the lack of a chapter on methods for determining the MIC value for individual substances. Summaries (for example in the form of a table) of how the authors of the publications from which you derived this review determined the MIC values were very interesting and would add value to the manuscript.
Response: We think this is an excellent suggestion. As suggested by the reviewer, we have added a new table summarizing the methods for the determination of MIC of antimicrobial drugs against P. insidiosum.
Comment 2: In the described cases, the potential sources of the pathogen were identified? As pythiosis happens in animals, could they be a vector?
Response: Thank you for pointing out this interesting aspect of whether infected animals act as vectors in transmitting disease between animals or from animals to humans. The habitat of Pythium spp. including Pythium insidiosum includes moist soil and stagnant fresh water. So, these habitats should be the potential sources. In the case of animals as vectors, it has been reported that fungus gnats and shore flies could be the vector of Pythium spp. (excluding P. insidiosum) causing plant diseases. However, in P. insidiosum, the animal has not been reported as a vector. There is only one publication that mentioned the successful isolation of P. insidiosum from mosquito larvae (Vilela, R.; Montalva, C.; Luz, C.; Humber, R.A.; Mendoza, L. Pythium insidiosum isolated from infected mosquito larvae in central Brazil. Acta Trop 2018, 185, 344-348, doi:10.1016/j.actatropica.2018.06.014).
Comment 3: Please see the requirements for citing works by the journal. It is different from what you have in your manuscript.
Response: We have formatted the references in the ACS style as recommended by the journal.
Comment 4: Please format the text and table as required by the journal.
Response: We have inserted tables into the main text close to their first citation
Reviewer 3 Report
Thanks for the interesting topic. Here are my suggestions to consider:
Title: Antimicrobial therapy for human pythiosis: A review
- Consider including clinical photos for human pythosis
- Line 30: Please provide a reference citation
- Line 102: Give an example of PK "issues" that may compromise the effectiveness of antimicrobial treatment
- Consider eliminating redundancy and unnecessary repetition
Author Response
Thank you for taking the time to assess our manuscript. We appreciate the positive feedback from the reviewer.
Comment: Consider including clinical photos for human pythiosis
Response: As suggested by the reviewer, we have included a clinical photo for human pythiosis (with permission) (See Figure 1).
Comment: Line 30: Please provide a reference citation
Response: Thank you for pointing this out. We have added the relevant reference in the revised manuscript.
Comment: Line 102: Give an example of PK "issues" that may compromise the effectiveness of antimicrobial treatment
Response: We think this is an excellent suggestion. As suggested by the reviewer, we have added examples of PK issues that compromise the safety and efficacy of antimicrobial treatment (See lines 141-153).
Comment: Consider eliminating redundancy and unnecessary repetition
Response: As suggested by the reviewer, we have significantly tried to avoid redundancies in the revised manuscript.
Reviewer 4 Report
The abstract should be rewritten to better describe the purpose, methodology and results of the study
The bibliography should be added to Figure 1.
In the discussion, it should be considered that this infection is not present in the Sanford Guide
Author Response
Dear Reviewer,
We want to thank the reviewer for a thorough reading of our manuscript. We appreciate the constructive suggestions from the reviewer.
Comment: The abstract should be rewritten to better describe the purpose, methodology and results of the study
Response: As suggested by the reviewer, we have rewritten the abstract more structured way.
Comment: The bibliography should be added to Figure 1.
Response: Thank you for pointing this out. We have added the relevant references to now retitled Figure 2.
Comment: In the discussion, it should be considered that this infection is not present in the Sanford Guide
Response: As suggested by the reviewer, we have added the information regarding the absence of human pythiosis in the Sanford Guide (See lines 577-579).
Sincerely,
Dr. Navaporn Worasilchai
Round 2
Reviewer 3 Report
Thanks for implementing the changes and including a clinical photo